# A Comparison of Mother’s Milk and the Neonatal Urine Metabolome: A Unique Fingerprinting for Different Nutritional Phenotypes

**DOI:** 10.3390/metabo12020113

**Published:** 2022-01-25

**Authors:** Angelica Dessì, Alessandra Marzullo, Sara Corbu, Alice Bosco, Flaminia Cesare Marincola, Maria Grazia Pattumelli, Michele Mussap, Roberta Pintus, Vassilios Fanos, Rocco Agostino

**Affiliations:** 1Departement of Surgical Science, University of Cagliari and Neonatal Intensive Care Unit, AOU Cagliari, 09124 Cagliari, Italy; alicebosco88@gmail.com (A.B.); mumike153@gmail.com (M.M.); gomberta@icloud.com (R.P.); vafanos@tiscali.it (V.F.); 2Ospedale San Giovanni Calibita, Fatebenefratelli, 00186 Roma, Italy; marzullo.alessandra@gmail.com (A.M.); mpattumelli@gmail.com (M.G.P.); rocco.agostino@uniroma1.it (R.A.); 3Department of Chemical and Geological Sciences, University of Cagliari, Cittadella Universitaria, SS 554, km 4.5, Monserrato, 09042 Cagliari, Italy; sari.crb@gmail.com (S.C.); flaminia@unica.it (F.C.M.)

**Keywords:** maternal milk, metabolomics, human milk oligosaccharides, urine metabolome, milk metabolome

## Abstract

The ability of metabolomics to provide a snapshot of an individual’s metabolic state makes it a very useful technique in neonatology for investigating the complex relationship between nutrition and the state of health of the newborn. Through an ^1^H-NMR metabolomics analysis, we aimed to investigate the metabolic profile of newborns by analyzing both urine and milk samples in relation to the birth weight of neonates classified as AGA (adequate for the gestational age, *n* = 51), IUGR (intrauterine growth restriction, *n* = 14), and LGA (large for gestational age, *n* = 15). Samples were collected at 7 ± 2 days after delivery. Of these infants, 42 were exclusively breastfed, while 38 received mixed feeding with a variable amount of commercial infant formula (less than 40%) in addition to breast milk. We observed a urinary spectral pattern for oligosaccharides very close to that of the corresponding mother’s milk in the case of exclusively breastfed infants, thus mirroring the maternal phenotype. The absence of this good match between the infant urine and human milk spectra in the case of mixed-fed infants could be reasonably ascribed to the use of a variable amount of commercial infant formulas (under 40%) added to breast milk. Furthermore, our findings did not evidence any significant differences in the spectral profiles in terms of the neonatal customize centile, i.e., AGA (adequate for gestational age), LGA (large for gestational age), or IGUR (intrauterine growth restriction). It is reasonable to assume that maternal human milk oligosaccharide (HMO) production is not or is only minimally influenced by the fetal growth conditions for unknown reasons. This hypothesis may be supported by our metabolomics-based results, confirming once again the importance of this approach in the neonatal field.

## 1. Introduction

Breast milk represents the gold standard for infant nutrition, not only for its macronutrient content but also for the high presence of bioactive compounds [1]. Its composition is extremely dynamic, yet it perfectly adheres to the nutritional needs of the newborn in order to guarantee a healthy development. Despite some inter-individual and temporal variations of macronutrient composition, breastmilk tends to be well preserved among populations, regardless of the nutritional status of the mother [1]. Above all, the presence of hundreds to thousands of different bioactive molecules provides the breast milk with many properties, including protection from infection and inflammation, and contribution to the maturation of the immune system and to the development of the nascent microbiota. Some of these compounds are characterized by a considerable variability, both between mothers and according to gestational age and the stage of breastfeeding, in support of an important reactivity of breast milk to the changing needs of the newborn [1]. Among these, the most represented are human milk oligosaccharides (HMOs), the third soluble component of milk, whose concentration varies considerably during lactation, decreasing from colostrum to mature milk [2]. Furthermore, the qualitative and quantitative composition of HMOs is strongly influenced by the maternal phenotype [2], diet [3], the pre-pregnancy BMI [4,5], the geographical area where the mother lives [6], and by several other environmental factors [7]. HMOs have many biological functions. Among these, there is an important modulation of the composition of the intestinal microbiota, whose influence affects numerous physiological processes in the newborn, such as the maturation of the intestinal mucosal barrier. Furthermore, the contribution of microbiota to some significant neonatal pathologies, such as necrotizing enterocolitis (NEC), is noteworthy. Brain development and the programming of the immune system are also strongly influenced by the presence of HMOs, with positive effects, especially in decreasing allergies and the incidence of autoimmune diseases [2].

The ability of metabolomics to provide a snapshot of an individual’s metabolic state makes it a very useful technique in neonatology for investigating the complex relationship between nutrition and the state of newborn health [8,9]. In fact, today it is clear that nutrition in the neonatal period can affect the future metabolic health of children and may prevent numerous chronic diseases. Therefore, the early identification of the most vulnerable subjects by introducing the personalized medicine model at the early stages of development would be desirable [8]. This is the case with newborns who are classified as large for gestational age (LGA), and those who have undergone intrauterine growth restriction (IUGR) [9]. In fact, both groups, regardless of their different intrauterine growth conditions, show a reduced tolerance to carbohydrates at birth, which tends to persist during growth and in adulthood, exposing them to a greater risk of metabolic pathologies [10].

In this metabolomics-based study, we aimed to investigate the metabolic profile of newborns by analyzing both urine and milk samples in relation to birth weight, AGA (adequate for gestational age), LGA and IUGR.

## 2. Results and Discussion

Overall, 80 mothers provided a human milk sample collected at 7 ± 2 days post-partum, and a urine specimen was taken from their own infant on the same day. Forty-two infants received exclusively their own mother’s milk, while 38 received a mixed feeding with a variable amount of commercial infant formulas (under 40%) in addition to breast milk.

A representative ^1^H NMR spectrum of human milk is depicted in Appendix A. Based on data from the literature [6,11,12], the analysis of the HMO regions of the milk ^1^H NMR spectrum provided information on maternal phenotypes. The fucosylation pattern of human milk oligosaccharides (HMOs) mostly reflects the expression and activity of two genes located in two different loci of the short arm of chromosome 19: the secretor gene (Se) and the Lewis gene of blood groups (Le). The first gene codes for the α1,2-fucosyltransferases, while the latter codes for the α1,3- /α1,4-fucosyltransferases. Accordingly, the presence and the absence of HMO peaks in the milk spectra under investigation allowed us to identify four different mother phenotypes: Se^+^/Le^+^ (*n* = 51), Se^−^/Le^+^ (*n* = 17), Se^+^/Le^−^ (*n* = 10), and Se^−^/Le^−^ (*n* = 2). As reported in Figure 1A, the milk spectrum of mothers classified as Se^+^/Le^+^ exhibited the signals of α 1,2-linked fucosylated HMOs, such as 2′fucosyllactose (2′FL; *δ* 4.53, *δ* 5.32) and lactodifucotetraose (LDFT; *δ* 1.27), and the peak representative of α 1,4-linked fucosylated HMOs, such as lacto-*N*-fucopentaose II (LNFP II, *δ* 5.03). In contrast, the spectrum of Se^+^/Le^−^ mothers (Figure 1B) was characterized by the absence of peaks from α1,4-fucosylated HMOs, those with the Se^−^/Le^+^ phenotype (Figure 1C) lacked signals from α1,2-fucosylated structures, while the spectrum of Se^−^/Le^−^ mothers (Figure 1D) exhibited only the signals of α1,3-fucosylated oligosaccharides.

**Figure 1 metabolites-12-00113-f001:**
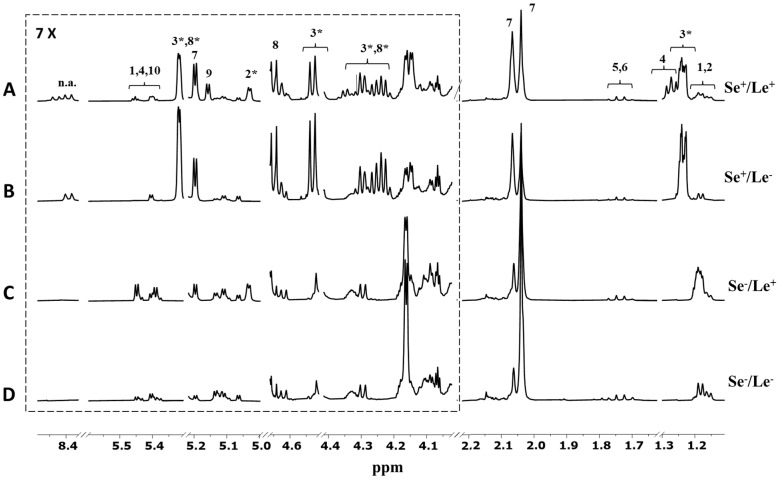
Expansions of the HMO NMR spectral regions (500 MH, 300 K) of four milk samples belonging to (**A**) Se^+^/Le^+^, (**B**) Se^+^/Le^−^, (**C**) Se^−^/Le^+^, and (**D**) Se^−^/Le^−^ phenotypes. Peak intensities were normalized to the sum of the total spectral area. Spectral attribution was performed according to the literature [11,12,13,14]. Key: 1—3fucosyllactose (3FL); 2—lacto-*N*-fucopentaose II (LNFP II); 3—2′fucosyllactose (2′FL); 4—lactodifucotetraose (LDFT); 5—3′ sialyllactose; 6—6′ sialyllactose; 7—*N*-acetylglucosamine; 8—lacto-*N*-fucopentaose I (LNFP I); 9—lacto-*N*-fucopentaose III (LNFP III); 10—lacto-*N*-difucohesaose II (LNDFH II); n.a.—not assigned. The asterisk denotes the peaks used for mother phenotype assignment.

The overall variability of the milk data set was explored by principal component analysis (PCA). The score distribution in the PC1 vs. PC2 plot did not suggest any difference in the spectral profiles in terms of the neonatal customize centile, i.e., AGA, LGA, or IUGR (Figure 2A). Otherwise, noticeable sample clustering along the PC1 axis between the spectral profiles of secretors (Se^+^/Le^+^, Se^+^/Le^−^) and non-secretors (Se^−^/Le^+^, Se^−^/Le^−^) was visible [15,16,17,18]. Additionally, a slight separation was observed within the secretor group in terms of Lewis status (i.e., Se^+^/Le^+^ and Se^+^/Le^−^) (Figure 2B).

The orthogonal partial least squares discriminant analysis (OPLS-DA) pairwise comparisons (Figure 3A,B) between Se^+^ and Se^−^ and between Le^+^ and Le^−^ confirmed HMOs as the metabolites mainly contributing to the class separation: α1,2-fucosylated structures, such as 2′FL, LNFPI, LDFT, and LNDFHI, characterized the Se^+^ group, while oligosaccharides with α1,3-fucosyl residues (3FL, LNFP II, LDFT, and LNDFH II) were found to be more abundant in the Se^−^ group (Figure 3A). A discrete discrimination and predictive capability characterized the model built for the Le^+^ and Le^−^ comparison (R^2^Y = 0.843; Q^2^ = 0.411; *p* < 0.001, Figure 3C,D) confirming the α1,4-fucosylated structures (LNFP II) as the key metabolites in the discrimination between the two Lewis status’.

The HMO spectral pattern of infant urine samples was also analyzed and compared to those of the corresponding mother’s milk. Due to the marked overlapping of HMO peaks in urine with those arising from other metabolites, only the spectral region between 5.0 and 5.5 ppm was considered (Figure 4). A good match between the HMO spectra region of infant urine and mother’s milk was observed for the infants who were exclusively breastfed. The absence of a urinary HMO pattern mirroring the maternal phenotype in the case of mixed-fed infants could be reasonably ascribed to the use of a variable amount of commercial infant formulas (under 40%) in addition to breast milk. Urine specimens from infants of secretor and non-secretor mothers were clearly distinguished by the presence or the absence of the α1,2-fucose residues peak at 5.32 ppm from 2′FL and LNFPI. In contrast, the identification of Lewis status was not always reliable because of the low intensity of the peaks from α1,4-fucosyl moieties (5.04 ppm).

The PCA model, built with the urine data set, did not evidence any sample separations in terms of neonatal centile and milk phenotype (Appendix A), whereas a clustering was observed according to the type of feeding (Figure 5A). This result is not surprising, since early feeding is known to play an important role on the composition of the urine metabolome [19]. These findings are somewhat similar to those previously found by Dessì et al. in a cohort of neonates categorized according to their weight at birth (LGA, IUGR, and AGA) and monitored during a seven-day timeframe (one week of postnatal life) [9]. In that study, differences in the neonatal metabolism depended only on the type of feeding (exclusive breastfeeding, mixed feeding, or exclusive formula). To assess the metabolic fingerprint differentiating infants who were exclusively breastfed from those who received a mixed feeding, an OPLS-DA model was built in pairwise comparison. Despite the presence of differences in the levels of various metabolites (such as higher levels of citrate, creatinine, and urea in the group of exclusively breastfed infants, and higher levels of lactate, glycine, and maltodextrin in the mixed-fed group), the descriptive and predictive power of the model was low (R^2^Y = 0.639, Q^2^ = 0.216, *p* < 0.001; Figure 5B,C).

Creatinine is the waste product of muscle metabolism, and is excreted in the urine by glomerular filtration at a relatively constant rate. Creatinine is formed from the degradation of creatine, a key molecule for the energy metabolism of the muscle. In the pediatric population, both creatine and muscle mass increase in a time-dependent manner with growth, and with the progressive greater movements during the child’s development [20]. However, in the first weeks of life there is a decrease in urinary creatinine level, reflecting its serum decrease. This may depend on the immaturity of the renal function of newborns (lack of passive tubular back diffusion) [21,22], or even on residual maternal creatinine, especially in the first days of life. Therefore, a certain instability of creatinine serum and urinary values is commonly observed during the first week of life. Data from the literature indicate that the creatinine steady state is reached approximately within the first month of life [23]. In a study performed by Marincola et al. [24], higher creatinine levels were found at birth in the urine of IUGR and LGA infants, supporting the metabolic alterations that characterize these infants. Seven days later, this alteration persisted only in LGA infants. On the other hand, in our study, exclusively breastfed infants, regardless of being born as LGA or IUGR, are characterized by higher levels of not only creatinine but also urea, the main nitrogen waste product resulting from protein degradation. However, the instability of creatinine values at seven days from birth does not make it a reliable metabolite, and the high urea values in exclusively breastfed infants could suggest a different growth pattern compare to that in babies who received the addition of formula milk, despite the lower protein intake [25,26,27].

Concerning lactate, we found higher levels of this metabolite in mixed-fed babies according to the pilot study performed by Shoji H et al. [28]. However, that study compared a group of healthy children with maternal-prevalent mixed breastfeeding (<540 mL/week of their intake from formula) and exclusively artificial breastfeeding at 1 and 6 months of life. Higher lactate levels were found in children fed exclusively with formula milk compared with the other group. Consequently, it was hypothesized that the increase was due to the higher concentrations of post-prandial insulin, a peculiar hormonal response exclusively recognizable in formula-fed infants and promoted by greater glucose absorption into cells [29]. The authors of the study concluded that the type of breastfeeding probably affects the metabolism of lactate at both 1 and 6 months of life [28]. In our study, the higher lactate values in children with mixed feeding may be attributed to the metabolic impact of the formula, albeit present in a small quantity.

Citrate is an intermediate of the tricarboxylic acid (TCA) cycle that is involved in several metabolic pathways, including the regulation of insulin secretion in response to glucose and the modulation of lipid synthesis [24,30]. It is therefore influenced by various factors, especially diet and acid/base balance. However, in the pediatric population, it was found to be an age-related metabolite, even if its urinary excretion has not yet been fully clarified [19]. On the one hand, some metabolomics-based studies have highlighted that citrate seems to decrease with age [20,31], confirming previous results obtained in the pediatric population [32]. On the other hand, these data are in contrast with the incremental rates of citraturia observed in the study by Kirejczyk et al. [33], who analyzed the trend of urinary citrate secretion in a population aged 2–18 years, and with results published by Cesare Marincola et al. [19]. Despite the weakness of the model, we might suppose that the high urinary citrate levels observed in infants fed exclusively with breast milk, regardless of being born LGA, IUGR, or AGA, are caused by different metabolic pathways arising from the type of nutrition. Indeed, similar results were found by Dessi et al. [9], who conducted a study based on the gas chromatography-mass spectrometry (GC-MS) metabolomic analysis of urine samples collected at different time points (birth, 3 and 7 days of life). The authors demonstrated that citric acid is one of the most significant metabolites discriminating breastfed babies on the seventh day of life. Even the study performed by Cesare Marincola et al. [19] found that the excretion of citrate was greater in breastfed infants than in those fed with formula. Furthermore, in another study performed using proton nuclear magnetic resonance (^1^H-NMR) spectroscopy [24], urine citrate levels were higher in IUGR and LGA newborns compared to AGA newborns, with incremental values from birth to the seventh day of life. It could be assumed that when the influence of nutrition is not yet important [9], the high concentration of citrate in the urine at birth for IUGR and LGA may result from fetal malnutrition, as emerged from a recent study by Bahado-Singh et al. [34]. By coupling metabolomics with artificial intelligence, the authors observed various alterations in the energy generation pathways in newborns with IUGR. In detail, at the level of ß-pancreatic cells, given the absence of gluconeogenesis and the low rate of lipid synthesis, some TCA cycle intermediates, such as citric acid, leave the mitochondria and travel to the cytosol, where they can act as cell signals for glucose-dependent insulin secretion [30]. As postnatal age increases, the metabolic impact of diet becomes preponderant, and the high citrate concentration found in exclusively breastfed infants may indicate the prevalence of glucose and lipid catabolism over that of proteins [35], taking into account the relatively low protein content in breast milk compared to that in formulas. Alterations in the urinary concentration of several TCA cycle intermediates, namely cis-aconitate, formate, and lactate [19], seem to confirm the hypothesis on the close association between the type of diet and the activation of alternative pathways for energy supply.

Glycine is a non-essential amino acid involved in the synthesis of glutathione (γ-glutamyl-cystenyl-glycine; GSH), one of the most important intracellular antioxidant systems in mammalian cells. GSH represents a time-dependent metabolite, and its concentration decreases over the first year of life, probably due to its extensive use in the synthesis of muscle tissue during growth, similar to creatinine [20]. Glycine is essential for the synthesis of heme, purines, collagen, and elastin [36]. A decrease in glycine availability may be due to protein malnutrition [37], sepsis, or an excessive inflammatory response, or also to high liver oxidation induced, for example, by diabetes or high glucagon concentrations [20,38]. According to the results reported by Dessì et al. [9], it is reasonable to assume that glycine characterizes both IUGR and LGA newborns at birth, supporting an altered metabolic condition with catabolic predominance and a potential strong oxidative stress. These observations are in agreement with those reported by Marincola et al. [24], and are also correlated with the alterations in liver function detected in IUGR children through metabolomic analysis coupled with artificial intelligence [36]. Moreover, in the study by Dessì et al. [9], it was found that at the seventh day of life, under the major influence of the nutritional regimen, glycine becomes a discriminant metabolite for formula-fed babies. This result is in agreement with what was observed in our analysis, where higher levels of glycine were found in children with mixed nutrition. The correlation between the intake of formula and high levels of glycine could be determined by a lower metabolic efficiency or greater oxidative stress.

To explore the difference between the metabolic profiles of infants born from secretor and non-secretor mothers, an OPLS-DA approach was also applied to the whole urine metabolic data set. Although this model was not as robust as that obtained for the milk samples, it was able to separate the two infant groups with a discrete goodness of fit and predictability as indicated by an R^2^Y value of 0.804 and a Q^2^ value of 0.375, respectively, with a cross-validation ANOVA (CV-ANOVA) *p*-value < 0.001. The results confirmed the α1,2-fucose moieties (2′FL and LNFPI) as the most important metabolites discriminating the two phenotypes (Figure 6).

It could be hypothesized that since mixed-fed babies are receiving a smaller dose of maternal milk, the HMO utilization by the gut microbiota of these glycans may be different, with a different protective effect or functions. This is in agreement with a recent study by He et al. [39] conducted on the fecal microbiome and metabolome of children subjected to different types of breastfeeding. In infants fed with formula milk, the authors observed that the presence of low and different levels of oligosaccharides together with high protein levels affected the difference between the microbiome and the fecal metabolome in breastfed infants and those in formula-fed infants. The authors speculated that the possible cause may be a different microbial fermentation of carbohydrates and proteins as energy substrates.

In conclusion, our results showed that at seven days after birth, there were no differences in the urinary metabolome of newborns in terms of different fetal growth conditions, supporting the great metabolic impact of the type of feeding of the child in this phase of development [9]. Furthermore, the same HMOs produced by mothers were found in the urine of newborns, but only if they were exclusively breastfed. This data is in agreement with the fact that the degree of exclusivity of breastfeeding influences the development of the intestinal microbiota [23,40]. Furthermore, it confirms the importance of HMOs in modulating the microbiota itself. In this context, the metabolomic analysis performed on the urine of newborns provides some new research opportunities. In fact, the presence of specific metabolites at seven days of life that discriminate the type of breastfeeding regardless of the growth status of the newborns underlines the strong influence of nutrition from the first week of life. Furthermore, the presence in the urine of some metabolites related to metabolic alterations at birth, such as IUGR or LGA, determining the separation of newborns into two groups according to the degree of exclusivity of breastfeeding represents an important starting point for reflection on nutritional adequacy during the early stages of development.

In addition, the investigation of the urinary metabolome of newborns could provide some preliminary information about the presence of particular HMOs that are thought to be protective against some pathologies, such as necrotizing enterocolitis (NEC) and obesity. For instance, 2′-FL seems to attenuate NEC inflammation in humans [41] and LNFPI inversely correlates with infant weight and fat mass [2,42].

Further metabolomic investigations are certainly necessary in order to understand the specific metabolic pathways altered by the different types of feeding more precisely, to optimize formulas for infants, especially in the most susceptible subjects (e.g., IUGR, LGA), and to reduce the adverse outcomes associated with their unfavorable conditions with a view to increasingly personalized nutrition as a basis for preventive medicine [10,43].

## 3. Materials and Methods

### 3.1. Subject and Sample Collection

This study was performed in accordance with the Helsinki Declaration as revised in 1983. Written informed consent was obtained from the mothers before participation in the study. Milk collection was performed in accordance with the ethical standards of the responsible committee on human experimentation.

Human milk samples were obtained from 80 mothers of AGA, IUGR, and LGA term infants with a mean gestational age of 39 ± 3 weeks at 7 ± 2 days post-partum. Urine samples were also collected from the infants at the same time of milk sampling. Forty-two infants exclusively received their own mother’s milk, while 38 received a mixed feeding with a variable amount of commercial infant formulas (under 40%) in addition to breast milk. In this case, the formula milk was added at the end of breastfeeding for a maximum of two daily intakes.

The information regarding the subjects and samples is summarized in Table 1. The exclusion criteria were as follows: major malformations, congenital heart diseases, perinatal asphyxia, sepsis, and need for surgery.

Milk samples were collected once per day (2–3 mL), always in the morning, at the same time and before breastfeeding, by directly pressing the breast in a sterile container. Infant urine samples (1 mL) were collected using a non-invasive method with a cotton ball inserted into a disposable diaper, aspired with a syringe, and transferred to sterile 1.5 mL vials. All biological specimens were stored at −80 °C until NMR analysis.

### 3.2. Sample Preparation

On the day of ^1^H NMR analysis, both urine and milk samples were thawed at room temperature. Residual lipids and proteins were removed from the milk by centrifuging samples at 10,000× *g* for 30 min at 4 °C using Amicon Ultra 0.5 mL 10 kDa spin filters (Millipore, Billerica, MA, USA). Each filtered sample (350 μL) was mixed with 350 μL of phosphate buffer solution in D_2_O (0.1 M, pH 7.4) containing sodium 3-trimethylsilyl-(2,2,3,3-^2^H_4_)-1-propionate (TSP; final concentration 2 mM) and then transferred into a 5 mm wide NMR tube.

An aliquot of 8 μL of a 1% aqueous solution of NaN_3_ (an antibacterial) was added to 800 μL of urine. After centrifuging the samples at 12,000× *g* for 10 min at 4 °C to remove any solid particles, 630 μL of the supernatant solution was mixed with 70 μL of phosphate buffer solution in D_2_O (1.5 M, pH 7.4) containing TSP (final concentration 1 mM). The mixture was vortexed, and 650 μL was transferred into a 5 mm wide NMR tube.

### 3.3. NMR Spectroscopy

^1^H NMR experiments were performed at 300 K with a Varian UNITY INOVA 500 spectrometer (Agilent Technologies, Inc., Santa Clara, CA, USA), operating at a frequency of 499.83 MHz. One-dimensional (1D) ^1^H NMR spectra were acquired using a standard pulse sequence (1D NOESY), with pre-saturation during relaxation and mixing time for water suppression. Both the milk and urine spectra were acquired with 128 scans with 64 k data points, a spectral width of 6000 Hz, a recycle time of 3.5 s, and a mixing time of 1 ms. A 0.3 Hz line-broadening factor was applied to each spectrum prior to Fourier transformation. After phased and baseline correction, the chemical shift scale was set by assigning a value of δ = 0.00 ppm to the signal for the internal standard TSP.

### 3.4. Spectral Processing and Statistical Analysis

NMR spectra were processed using the MestreNova software, version 14.0.1 (Mestrelab Research SL, Santiago de Compostela, Spain) and corrected for misalignments in the chemical shift primarily due to pH-dependent signals. Each spectrum was segmented into 0.001 ppm integral regions (bins) between 9.5 and 0.5 ppm, excluding the portions with the residual water (δ 4.6÷5.2) and TSP resonances. The bins were normalized to the sum of the total spectral area to compensate for the overall concentration differences. The two final data sets were automatically reduced to ASCII files, converted into an Excel file, and then imported into the SIMCA 16.1 software (Umetrics, Umea, Sweden) where they were Pareto scaled. Principal component analysis (PCA) and orthogonal partial least squares discriminant analysis (OPLS-DA) were applied. The robustness of the OPLS-DA models was assessed based on the fitness (R_2_Y) and prediction (Q_2_Y) abilities determined through 7-fold cross-validation method. The calculation of CV-ANOVA [44] and y-table permutation testing over 400 iterations were additionally used to validate the models.

## Figures and Tables

**Figure 2 metabolites-12-00113-f002:**
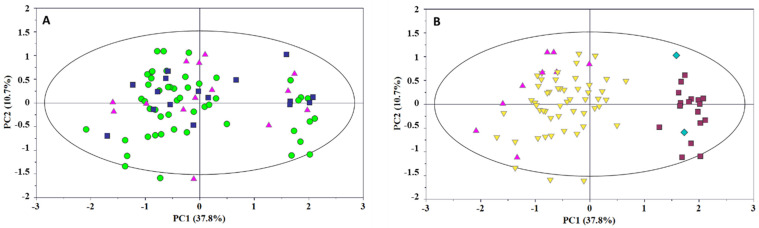
PC1 vs. PC2 scores plot of milk samples. Scores are colored according to: (**A**) the infant centile classification (AGA—green circle; LGA—blue square; SGA—pink triangle); (**B**) the milk phenotype (Se^+^/Le^+^—yellow inverted triangle; Se^+^/Le^−^—pink triangle; Se^−^/Le^+^—dark red box; Se^−^/Le^−^—blue diamond).

**Figure 3 metabolites-12-00113-f003:**
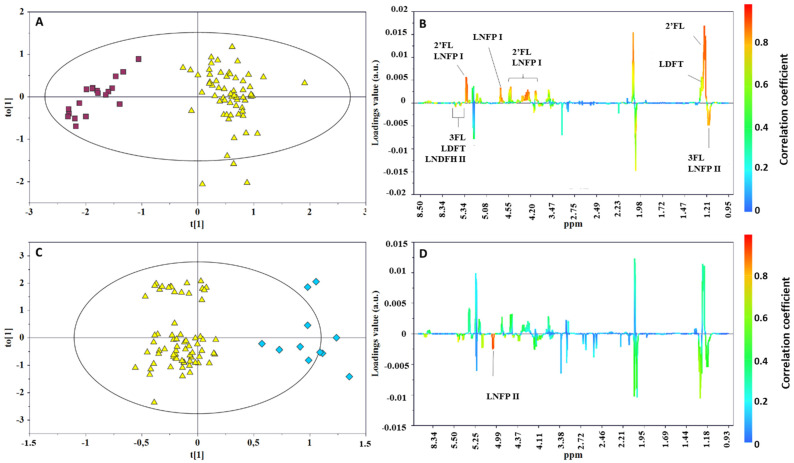
OPLS-DA scores (left) and correlation coefficient S-line (right) plots for the pairwise comparison of milk samples according to mother phenotypes: (**A**,**B**) Se^+^ (yellow triangle) vs. Se^−^ (red box). R^2^Y = 0.894, Q^2^ = 0.850, *p* < 0.0001; (**C**,**D**) Le^+^ (yellow triangle) vs. Le^−^ (blue diamond). R^2^Y = 0.843; Q^2^ = 0.411; *p* < 0.001).

**Figure 4 metabolites-12-00113-f004:**
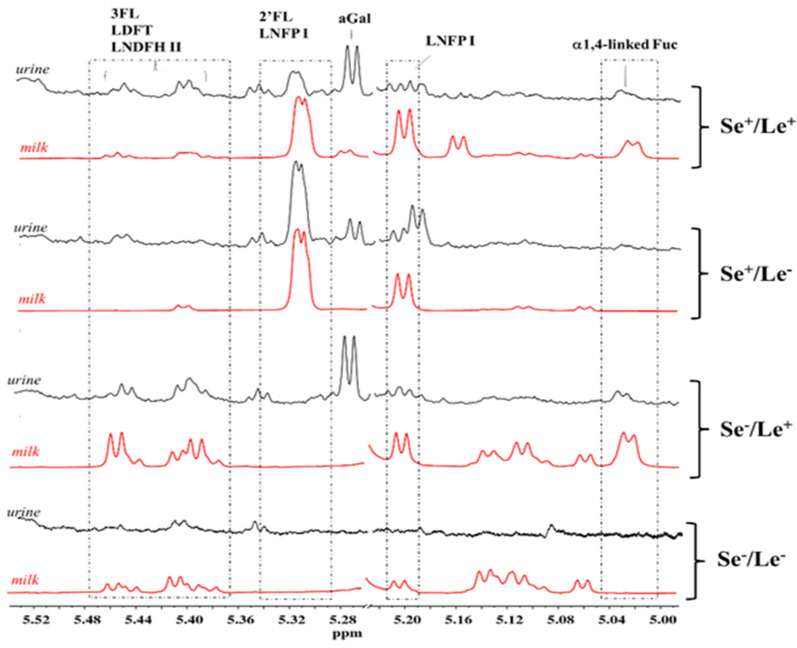
HMO NMR spectral regions between 5.0 and 5.5 ppm of four infant urine samples (black) and milk (red) samples associated to the Se^+^/Le^+^, Se^+^/Le^−^, Se^−^/Le^+^, and Se^−^/Le^−^mother’s phenotype. Abbreviations: 2′FL—2′fucosyllactose; 3FL—3fucosyllactose; LDFT—lactodifucotetraose; LNFP I—lacto-*N*-fucopentaose I; LNDFH II—lacto-*N*-difucohesaose II.

**Figure 5 metabolites-12-00113-f005:**
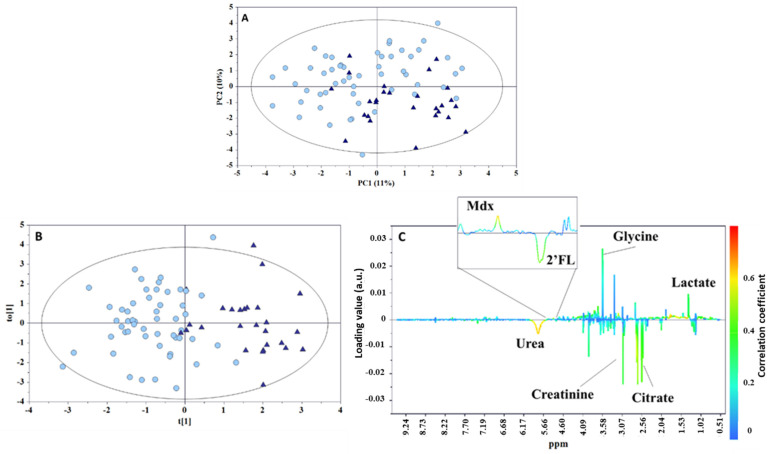
(**A**) PCA scores plot (PC1 vs. PC2) from ^1^H NMR infant urine spectra, (**B**) OPLS-DA scores, and (**C**) loadings plots for the two types of feeding groups: breastmilk (circle) and mixed feeding (triangle). R^2^Y = 0.639, Q^2^ = 0.216, *p* < 0.001. Mdx—maltodextrin; 2′FL—2′fucosyllactose.

**Figure 6 metabolites-12-00113-f006:**
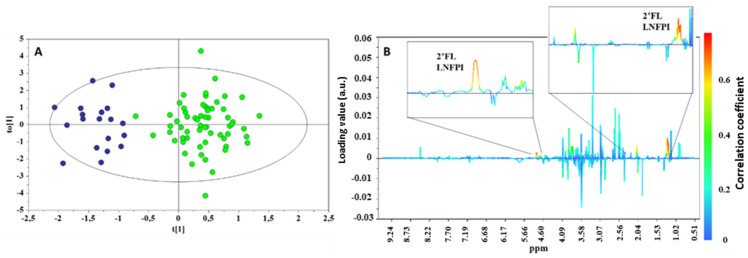
OPLS-DA scores (**A**) and correlation coefficient S-line (**B**) plots for the two groups of urine taken from infants born from a secretor (green) or a non-secretor (blue) mother. R^2^Y = 0.804, Q^2^ = 0.374, *p* < 0.001.

**Table 1 metabolites-12-00113-t001:** Maternal and infants’ characteristics.

	IUGR (*n* = 14)	AGA (*n* = 51)	LGA (*n* = 15)
**Maternal Characteristics**			
Age. years: (min; max)	34.71 (22.00; 44.00)	35.25 (24.00; 43.00)	34.13 (26.00; 44.00)
Type of delivery (vaginal delivery/cesarean section)	6/8	26/25	5/10
Amniocentesis	3	14	6
Chorionic villus sampling	2	1	0
**Infants characteristics at birth**			
Gender (male/female)	8/6	28/23	8/7
Gestational age, weeks (mean ± SD)	38 ± 4	39 ± 3	39 ± 2
Birth weight, g (min; max)	2412 (1450; 2790)	3236 (2.500; 4110)	3.975 (3220; 4490)
Length at birth, cm (min; max)	46.5 (30; 51)	50.1 (34.; 55)	50.3 (36; 57)
Head circumference, cm (min; max)	33.4 (32; 41)	35.7 (33; 51)	38.6 (34; 55)
Percentile	3.57°	46°	94.87°

## Data Availability

All data, tables and figures in this manuscript are original.

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
