# Peer review of "A Comparison of Mother’s Milk and the Neonatal Urine Metabolome: A Unique Fingerprinting for Different Nutritional Phenotypes"

_metabolites, 2022, doi:10.3390/metabo12020113_

Round 1

Reviewer 1 Report

The authors report the results of a study on the relationship between birth weight, breast or formula feeding and the metabolic profile of a group of newborns using the metabolomic technique on urine samples from newborns and milk from their mothers.  The study is very interesting and is part of a research field that could offer a lot of important health information. The great competence in the field of authors is very clear and it was a pleasure to read and comment on the article.

My comments are as follows:

Abstract. The text is clear and flowing and provides all the key information of the article.

Introduction. The text is exhaustive and the references adequate and updated.

The introduction is followed by the paragraphs "results and discussion" lines 76-288, while it should be followed by "materials and methods" lines 289-345. This is clearly a formal error but the paragraphs must be reversed.

Materials and methods

In many nurseries there is the custom of administering glucose solution to newborns. Did this also happen in this case? And do the authors believe that this practice could modify the urinary metabolomics of the newborn?

Line 299 on what basis was a percentage of formula administered less than 40% of the breast milk taken?

Line 302, were infants with sepsis that required the use of antibiotics excluded from the study?

Lines303-304 Was the milk sample taken at the beginning, in the middle or at the end of the feeding? I think it was taken at the beginning since it is the richest in sugar and water. Furthermore, was only one or more milk taken during the day?

Results and discussion

Line 125 last word "compare" should be "compared"

Line 131-132 have any differences been noted between the various formulas based on the possible different content of oligosaccharides? Has a gradient effect been noted for the amount of formula taken?

Line 133-134 Does the difference in Urine specimens from infants of secretor and non-secretor mothers have a clinical significance?

Line 179 The ref. n25 is old. Much more up-to-date data on the different growth of breast-fed or formula-fed infants is provided by:Gale, C.; Logan, K.; Santhakumaran, S.; Parkinson, J.R.C.; Hyde, M.J.; Modi, N. Effect of breastfeeding compared with formula feeding on infant body composition: a systematic review and meta-analysis. Am. J. Clin. Nutr. 2012, 95, 656-669; DOI:10.3945/ajcn.111.027284.   Breij, M.L.; Abrahamse-Berkeveld, M.; Acton, D. Impact of early infant growth, duration of breastfeeding and maternal factors on total body fat mass and visceral fat at 3 and 6 months of age. Ann. Nutr. Metab. 2017, 71(3-4), 203-210; DOI:10.1159/000481539.             

Line 236 “it is reasonable to assume it seems” it seems to me that authors have to decide which verb form should be used.

The conclusions lack a couple of summary sentences on the possible clinical use of these very interesting data.

Author Response

Dear Editor,

we want to thank all the reviewers for their precious suggestions to improve our work.

Reviewer 1

The authors report the results of a study on the relationship between birth weight, breast or formula feeding and the metabolic profile of a group of newborns using the metabolomic technique on urine samples from newborns and milk from their mothers.  The study is very interesting and is part of a research field that could offer a lot of important health information. The great competence in the field of authors is very clear and it was a pleasure to read and comment on the article.

My comments are as follows:

Abstract. The text is clear and flowing and provides all the key information of the article.

Thank you for this comment.

Introduction. The text is exhaustive and the references adequate and updated.

The introduction is followed by the paragraphs "results and discussion" lines 76-288, while it should be followed by "materials and methods" lines 289-345. This is clearly a formal error but the paragraphs must be reversed.

Thank you for this comment. The arrangement of the paragraphs was not our choice but reflects the requests of the Journal (Manuscript Preparation-General Considerations- Research manuscripts should comprise: Front matter: Title, Author list, Affiliations, Abstract, Keywords, Research manuscript sections: Introduction, Results, Discussion, Materials and Methods, Conclusions (optional),Back matter: Supplementary Materials, Acknowledgments, Author Contributions, Conflicts of Interest, References).

Materials and methods

In many nurseries there is the custom of administering glucose solution to newborns. Did this also happen in this case? And do the authors believe that this practice could modify the urinary metabolomics of the newborn?

Thanks for this question, no in this nursery there isn’t this custom. We can hypothesize that the administration of glucose solution by altering the energy metabolism can affect the urinary metabolome of the newborn.

Line 299 on what basis was a percentage of formula administered less than 40% of the breast milk taken?

Thanks for this comment. We have specified this detail in the study as following: ”the addition of formula milk in infants with mixed breastfeeding was reserved at the end of breastfeeding and for a maximum of two daily intakes”.

Line 302, were infants with sepsis that required the use of antibiotics excluded from the study?

Thanks for pointing out this inaccuracy.  We have corrected the exclusion criteria. Yes, infants with sepsis that required the use of antibiotics were excluded from the study.

Lines303-304 Was the milk sample taken at the beginning, in the middle or at the end of the feeding? I think it was taken at the beginning since it is the richest in sugar and water. Furthermore, was only one or more milk taken during the day?

Thank you for pointing out this criticality that we have better clarified in the text as following: “the collection of milk samples took place once per day, always in the morning, at the same time and before breastfeeding the baby”.

Results and discussion

Line 125 last word "compare" should be "compared"

Thank you for this comment, we have rectified this inaccuracy.

Line 131-132 have any differences been noted between the various formulas based on the possible different content of oligosaccharides? Has a gradient effect been noted for the amount of formula taken?

In the study, given the low consumption of formula milk, the different contributions of the various formulas analyzed were not tested.

Line 133-134 Does the difference in Urine specimens from infants of secretor and non-secretor mothers have a clinical significance?

Thank you for this question. Maternal genetic factors (secretor or non-secretor) correlates to different patterns of human milk oligosaccharides HMOs in human milk and given the importance of these compounds and their ability to influence the health of a neonate, the knowledge of a mother’s phenotype is fundamental for clinicians. Infact, it could be very important to know whether a particular human milk oligosaccharide that is thought to be protective against a pathology, like necrotizing enterocolitis (NEC) and obesity, is present in the milk of a mother or not, possibly supporting infants with adequate supplementation. In this context, the urinary metabolome of newborns could provide some preliminary information about the presence of 2’-fucosyllactose (2’-FL) and Lacto-N-Fucopentaose I (LNFPI). The first seems to attenuate NEC inflammation in humans (Sodhi CP, Wipf P, Yamaguchi Y, et al. The human milk oligosaccharides 2'-fucosyllactose and 6'-sialyllactose protect against the development of necrotizing enterocolitis by inhibiting toll-like receptor 4 signaling. Pediatr Res. 2021;89(1):91-101. doi:10.1038/s41390-020-0852-3) and the latter inversely correlates with infant weight and fat mass in the study of Alderete at al. (Alderete TL, Autran C, Brekke BE, Knight R, Bode L, Goran MI, Fields DA. Associations between human milk oligosaccharides and infant body composition in the first 6 mo of life. Am J Clin Nutr. 2015 Dec;102(6):1381-8. doi: 10.3945/ajcn.115.115451.)

Line 179 The ref. n25 is old. Much more up-to-date data on the different growth of breast-fed or formula-fed infants is provided by:Gale, C.; Logan, K.; Santhakumaran, S.; Parkinson, J.R.C.; Hyde, M.J.; Modi, N. Effect of breastfeeding compared with formula feeding on infant body composition: a systematic review and meta-analysis. Am. J. Clin. Nutr. 201295, 656-669; DOI:10.3945/ajcn.111.027284.   Breij, M.L.; Abrahamse-Berkeveld, M.; Acton, D. Impact of early infant growth, duration of breastfeeding and maternal factors on total body fat mass and visceral fat at 3 and 6 months of age. Ann. Nutr. Metab. 201771(3-4), 203-210; DOI:10.1159/000481539.   

Thank you for this comment and for suggesting this interesting studies. We have integrated these data in the manuscript and deleted this reference.

Line 236 “it is reasonable to assume it seems” it seems to me that authors have to decide which verb form should be used.

Thank you for this comment, we have rectified this inaccuracy by deleting it seems.

The conclusions lack a couple of summary sentences on the possible clinical use of these very interesting data.

Thank you for this comment and for suggesting this implementation. We have added it in the text.

Reviewer 2 Report

The authors conducted a comparative investigation on the metabolic 1H NMR spectral profiles of the two groups of infants’ urine (human-milk-fed infants vs mixed-milk-fed infants). The first half was profiling the spectral differences of HMOs in the human milk samples (Se+/- and Le+/­), followed by the different groups of infants’ urines. No significant differences were found with the neonatal centile; however, metabo discriminations were deduced from the urines between the groups.

Overall, the manuscript consists of a good introduction with a modest section results & discussion. In my opinion, a few extract spectral data (as supplementary) could be beneficial for clarification and facilitating the readers. 

Comments

-p2 lines 90, 92: typos – replace the fonts @

-p2 lines 86-96 describe the peak assignments discriminating the different HMOs in Figure 1. To facilitate the readers, authors should consider ‘marking or labeling’ these peak differences in Figure 1.

-Figure 1: are the peak intensity normalized to a specific variable (or peak)? For example, it seems that two of the Se- spectra have lower peak intensity (concentration) than those of Se+; if so, it would be an idea to address this in the manuscript

-Figures 2 & 3: Based on the description in the spectral processing for statistical analyses, each spectrum was binned between 0.5 – 9.5ppm excluding the water peak; as such, the authors should consider showing the entire 1H spectra of each of the four different HMOs as supporting materials, because Figure 1 seems to exclude some non-water peaks (e.g., at 4.4ppm and 5.2ppm and 2.5-4ppm, and more)

-Figure 3D: related to the above point, the peak assignment of LNDFHI-II at ca 5ppm is not shown in the corresponding spectra in Figure 1.

-Figure 4: should consider overlay with the different spectra of HMOs samples to facilitate the spectral comparison (maybe as supporting material). This would add readers’ confidence in the claim at p4, lines 125-128, and elsewhere in the text.

-it’s intriguing that no HMOs signatures (as claimed) are observed in the spectra of the urine extracted from the mixed milk-fed infants, even though 60% of mixed milk was breastmilk? One would expect there would be a trace of HMOs in the urine.

-had the author also collected NMR spectra of the commercial milk? Any significant (or worth noting) resonances from the mixed-fed infants ascribed from this commercial milk?

-p6 line 202-203: “In our analysis, infants fed exclusively with breast milk had higher urinary citrate levels than infants with mixed breastfeeding, regardless of being born LGA, IUGR or AGA.” where and how did you deduce this info, the loading of PCA in Figure 5 or the corresponding OPLS-DA? What is the correlation coefficient, P[corr]?

Author Response

Dear Editor,

we want to thank all the reviewers for their precious suggestions to improve our work.

Reviewer 2

The authors conducted a comparative investigation on the metabolic 1H NMR spectral profiles of the two groups of infants’ urine (human-milk-fed infants vs mixed-milk-fed infants). The first half was profiling the spectral differences of HMOs in the human milk samples (Se+/- and Le+/­), followed by the different groups of infants’ urines. No significant differences were found with the neonatal centile; however, metabo discriminations were deduced from the urines between the groups.

Overall, the manuscript consists of a good introduction with a modest section results & discussion. In my opinion, a few extract spectral data (as supplementary) could be beneficial for clarification and facilitating the readers. 

Comments

-p2 lines 90, 92: typos – replace the fonts @

Thank you for this comment, we have rectified this inaccuracy.

-p2 lines 86-96 describe the peak assignments discriminating the different HMOs in Figure 1. To facilitate the readers, authors should consider ‘marking or labeling’ these peak differences in Figure 1.

Following the reviewer suggestion, we have better detailed in the manuscript (lines 91-93 of the revised version) how the HMOs peak attribution allowed the assignment of mother phenotype. Furthermore, Figure 1 has been remade where HMOs peaks have been marked with a number. An asterisk has been also added to peaks used for the attribution of mother phenotype. These details have been added in the caption of Figure 1 (lines 100-103)

-Figure 1: are the peak intensity normalized to a specific variable (or peak)? For example, it seems that two of the Se- spectra have lower peak intensity (concentration) than those of Se+; if so, it would be an idea to address this in the manuscript

Figure 1 was prepared with no normalization of data. Then, what the reviewer noted is correct. We have redone Figure 1 normalizing peak intensities to the sum of the total spectral area and using a seven times magnification for the spectral region of HMOs peaks above 4 ppm

-Figures 2 & 3: Based on the description in the spectral processing for statistical analyses, each spectrum was binned between 0.5 – 9.5ppm excluding the water peak; as such, the authors should consider showing the entire 1H spectra of each of the four different HMOs as supporting materials, because Figure 1 seems to exclude some non-water peaks (e.g., at 4.4ppm and 5.2ppm and 2.5-4ppm, and more)

Following the suggestion of reviewer, we have added a new figure (Figure S1) as supporting material. Since the whole spectra of the three mother phenotypes are basically identical if plotted from 0.5 to 9 ppm, we have though that a representative 1H NMR spectrum of human milk, flanked by some expanded regions, could be more useful for a visualization of the whole spectral profile of milk.

Figure 3D: related to the above point, the peak assignment of LNDFHI-II at ca 5ppm is not shown in the corresponding spectra in Figure 1.

We thank the reviewer to this correct observation. There was a mistake in the figure 3D. The right name of the marked peak is LNFP II, in line with the peak attribution reported in Figure 1.

-Figure 4: should consider overlay with the different spectra of HMOs samples to facilitate the spectral comparison (maybe as supporting material). This would add readers’ confidence in the claim at p4, lines 125-128, and elsewhere in the text.

Based on the reviewer comment, we have decided to renew Figure 4 including also the NMR spectra of milk corresponding to each mother phenotype. We hope that this choice can improve the exposition of our findings.

-it’s intriguing that no HMOs signatures (as claimed) are observed in the spectra of the urine extracted from the mixed milk-fed infants, even though 60% of mixed milk was breastmilk? One would expect there would be a trace of HMOs in the urine.

Thank you for this comment. We might have expressed the concept unclearly. Our intention is to stress the possibility that the presence in artificial milk of lower and different oligosaccharides levels might affect the difference observed in the urinary metabolome because the HMOs utilization by the gut microbiota of these glycans may be different. This speculation is in agreement with a recent study by He et al. on the fecal microbiome and metabolome of children subjected to different types of breastfeeding.

-had the author also collected NMR spectra of the commercial milk? Any significant (or worth noting) resonances from the mixed-fed infants ascribed from this commercial milk?

Unfortunately, we did not. As highlighted by the reviewer, it would have been interesting to do it

-p6 line 202-203: “In our analysis, infants fed exclusively with breast milk had higher urinary citrate levels than infants with mixed breastfeeding, regardless of being born LGA, IUGR or AGA.” where and how did you deduce this info, the loading of PCA in Figure 5 or the corresponding OPLS-DA? What is the correlation coefficient, P[corr]?

Thank you for this comment, we have rectified this inaccuracy implementing Figure 5 to highlight what is expressed in the discussion. We have also specified in the text that “despite the weakness of the model we might suppose that the high urinary citrate levels observed in infants fed exclusively with breast milk, regardless of being born LGA, IUGR or AGA, are caused by different metabolic pathways arising from the type of nutrition”.